# Liver Organoids: Updates on Disease Modeling and Biomedical Applications

**DOI:** 10.3390/biology10090835

**Published:** 2021-08-27

**Authors:** Carmen Caiazza, Silvia Parisi, Massimiliano Caiazzo

**Affiliations:** 1Department of Molecular Medicine and Medical Biotechnology, University of Naples “Federico II”, Via Pansini 5, 80131 Naples, Italy; carmen.caiazza@unina.it; 2Department of Pharmaceutics, Utrecht Institute for Pharmaceutical Sciences (UIPS), Utrecht University, Universiteitsweg 99, 3584 CG Utrecht, The Netherlands

**Keywords:** 3D culture, hepatocyte, cholangiocytes, stem cells, liver disease modeling, personalized medicine

## Abstract

**Simple Summary:**

Liver organoids represent a breakthrough for disease modeling as proven by their ability to recapitulate pathophysiological morphology and functional features of the original tissue. This article reviews the works that established liver organoids and the recent improvements of this novel in vitro system. We discuss their application in biomedicine focusing on disease modeling, regenerative medicine and drug discovery.

**Abstract:**

Liver organoids are stem cell-derived 3D structures that are generated by liver differentiation signals in the presence of a supporting extracellular matrix. Liver organoids overcome low complexity grade of bidimensional culture and high costs of in vivo models thus representing a turning point for studying liver disease modeling. Liver organoids can be established from different sources as induced pluripotent stem cells (iPSCs), embryonic stem cells (ESCs), hepatoblasts and tissue-derived cells. This novel in vitro system represents an innovative tool to deeper understand the physiology and pathological mechanisms affecting the liver. In this review, we discuss the current advances in the field focusing on their application in modeling diseases, regenerative medicine and drug discovery.

## 1. Introduction

In vitro modeling is a powerful approach that allows the understanding of disease mechanisms in a human cellular system. However, due to the lack of cell-cell communication and tissue microenvironment, bi-dimensional (2D) cultures do not faithfully represent complete and accurate physiological and pathological features [1]. For this reason, since the 70’s, the development of three-dimensional (3D) cultures has been implemented [2,3]. In the last decades, several groups established 3D structures, defined organoids, obtained by culturing stem cells with an extracellular matrix (ECM), allowing a self-organization in a structure resembling structural and functional parameters of the tissue of interest [4,5]. The seminal work from Prof. Clevers’s lab demonstrated that a single leucine-rich repeat-containing G-protein-coupled receptor 5 (LGR5) stem cell [6] isolated from mouse intestinal tissue can form a functional 3D culture with crypt-villus architecture [7]. Since then, LGR5-positive (LGR5+) stem cells have been isolated from many other organs enlightening the pivotal role of Wnt signaling in the maintenance of the stemness [8].

## 2. Liver Organoids

Liver functions as endocrine and exocrine gland in bile production, plasma protein secretion, storage of macromolecules, metabolites production, hormone synthesis and drug metabolization. The complexity of liver role can be achieved by the coordinated interaction of different cell types such as hepatocytes, cholangiocytes and non-parenchymal cells (stellate cells, Kupffer cells, liver sinusoid endothelial cells and portal fibroblasts) [9]. The generation of liver organoids is ideal to represent in vitro such complexity, therefore enhancing our insight into human liver biology and pathology. Liver functionality can be assessed by measuring different parameters as metabolic activity, protein secretion, storage ability and detoxifying action. 2D and 3D hepatocyte cultures display a different behavior in maintaining such parameters. Bidimensional cultures of primary human hepatocytes rapidly lose proteins involved in metabolic pathways and express lower ability in detoxifying processes whereas 3D cultures were more functionally stable [10].

### 2.1. Adult Stem Cell-Derived Organoids

The first liver organoid system was achieved by culturing LGR5+ biliary cells derived from a liver injury mouse model in R-Spondin1 (Rspo1)-based medium [11]. The clonal organoids obtained could be cultured over several months harboring a normal karyotype and expressing both hepatocyte and bile duct markers revealing a bipotent differentiation. By inhibiting Notch and TGF-β signaling implied in biliary determination, Clevers’s lab demonstrated that the culture can be committed to hepatocyte differentiation. The derived population displayed the hepatocyte markers Hnf4-α, Albumin (Alb), Mrp4 and ZO-1. Moreover, the analysis of glycogen accumulation, LDL uptake and cytochrome p450 activity confirm the functionality of hepatocytes [11].

Two years later, the first 3D culture of human liver organoid was obtained by slightly adjusting the conditions used for murine organoids (Rspo1, EGF, FGF10, HFG, nicotinamide). By adding cAMP agonists to the medium, such as Forskolin (FSK) and the TGFβ receptor inhibitor A83.01, LGR5+ liver cells can be differentiated into liver organoids (Figure 1). When cultured in differentiation medium the organoids correctly differentiated into functional hepatocytes and were able to engraft in nude mice after liver injury-induced damage. Moreover, for the first time, a resembling disease phenotype liver organoid was obtained from patients with α-1 antitrypsin (A1AT) deficiency and Alagille syndrome (ALGS) [12].

Aside from the generation of cholangiocytes-derived organoids (chol-orgs) from EPCAM+ biliary epithelial cells, most recently a culture protocol for generating hepatocyte-derived organoids (hep-orgs) has been described (Figure 1). By increasing R-spondin and FGF signaling, Clevers’s group generated both a human and mouse hepatocyte-derived 3D culture with expression profiles that resemble that of hepatocytes after partial liver resection. Hep-orgs displayed a more accurate hepatocyte-like phenotype then chol-orgs as demonstrated by a higher expression of hepatocyte markers (Alb, Hnf4α, Cyp1a2, Cyp3a11) and a 1000-fold higher secretion of Albumin [13].

In parallel, a study conducted in Prof. Nusse’s lab demonstrated that the combination of growth factors and small molecules with TNF-α, a cytokine produced by Kupffer cells during liver injury, enhanced the expansion of primary hepatocytes in 3D culture while retaining the engraftment potential [14].

A recent work revealed that cell proliferation rate increased in human liver organoids upon the use of spinner flasks. The size of organoids at day 14 of seeding was five times bigger than the controls embedded in static Matrigel. mRNA-seq analysis revealed that the spinner flask derived organoids exhibited an upregulation of cell cycle genes and a downregulation of cell adhesion and hypoxia genes suggesting that a major disposal of oxygen could optimize the large-scale production of organoids for future therapeutic applications. Interestingly, the increase of growth rate did not result in the acquisition of a tumorigenic phenotype as demonstrated by the lack of neoplastic formation upon inoculation in mice. Besides, when cultured in differentiation medium, the spinner-flask organoids showed marker expression and functional activity more resembling mature hepatocytes compared the organoids obtained with Matrigel cultures. Nevertheless, the spinner flask organoids displayed inability to engraft in vivo, probably related to the downregulation of cell adhesion proteins [15].

### 2.2. Pluripotent Stem Cell-Derived Liver Organoids

Besides biopsies, another source for the generation of liver organoids is the patient-derived iPSCs (Figure 1). The first study that illustrated the enormous potential of iPSC-derived organoids was conducted by Prof. Taniguchi’s group in 2013. By co-culturing iPSC-derived endoderm (iPSC-HE) with human umbilical vein endothelial cells (HUVECs) and human mesenchymal stem cells (MSCs), the culture self-organized in a 3D iPSC-derived liver bud (iPSC-LB). This structure was able not only to engraft in NOD/SCID mouse model as proved by the secretion of human Albumin, but also to establish a vascularized transplant network connected to the host blood vessels [16]. The technique was further improved by translating the iPSC-HE production in a good manufacturing practice (GMP)-grade system for the future use in regenerative therapies. Septum transversum mesenchyme (STM)-IPSCs and endothelial progenitors (EC)-iPSCs were used as stromal cells for the generation of iPSC-LB without affecting the ability to engraft and to connect to the blood circulation [17]. Prof. Bazerre group showed the importance of paracrine signaling in liver organoid maturation by coculturing iPSC-HE with HUVEC and EC, thus proving that the role performed by stromal cells is both structural and functional. The cell-to-cell direct contact between the three cell lineages proved to be fundamental for the generation of a 3D structure whereas even in a separate co-culturing method (transwell system), the paracrine soluble factors induced iPSC hepatocytic differentiation as demonstrated by bile acid transport induction. Proteomics analysis identified angiotensin, α-2 macroglobulin and plasminogen as the paracrine factors that triggers hepatic differentiation [18].

A significant advance in the generation of liver organoids and in modeling human liver development arrived from the Peltz’s group. They developed a protocol to generate human hepatic organoids from iPSCs applying sequential changes in the growth factor and chemicals added to the culture and, thus, driving iPSC differentiation through stages that resemble human liver during its embryonic development [19]. These hepatic organoids contain 2 types of cells organized into a complex structure: sheets of hepatocytes and the cholangiocytes are organized into epithelia surrounding the lumina of bile duct–like structures. Of note, the authors reported that hepatic organoids showed a regenerative property that is characteristic of humans, indeed they are able to generate secondary organoids from a single primary organoid [19].

## 3. Xenofree Liver Organoids

One of the major challenges in the organoid field is the accomplishment of xenofree hydrogels for their culture. Currently, the organoids generation relies upon Matrigel, a gelatinous protein mixture derived from the ECM of the Engelbreth-Holm-Swarm mouse sarcoma that is mainly composed of laminin, collagen IV and entactin. This matrix is widely used because it supports the structure and proliferation of different organoid types and is extremely bioactive. However, Matrigel displays a series of disadvantages relying on both batch-to-batch variation that affect the reproducibility of organoids establishment and the unsuitability for clinical application [20]. For this reason, several studies have been conducted to find alternative synthetic hydrogels that permits the generation of organoids with the same efficiency of Matrigel without the possible immunogenic disadvantages. Promising results have been recently accomplished by using chemical derived matrices. Schoonjans’s group, employed a polyethylene glycol (PEG) hydrogel enzymatically crosslinked by the activated transglutaminase factor XIIIa (FXIIIa) [21]. This matrix allowed the growth of both mouse and human organoids from biliary ducts and non-tumorigenic needle biopsies [22]. Surprisingly, the generation of the organoids was accomplished without the use of animal components at any step of the process. The PEG matrix was complemented with key ECM proteins of native liver such as laminin-111, collagen IV and fibronectin. The replacement of fibronectin with its minimal integrin recognition peptide RGDSPG (Arg-Gly-Asp-Ser-Pro-Gly) did not affect the organoid generation. PEG-derived liver organoids were indistinguishable from Matrigel ones in terms of morphology and gene expression. Moreover, PEG-RGD gels allow the culture for more than 14 days whereas the growth in Matrigel is maintained only for 6 days. The preserved differentiation capacity of PEG-derived organoids was confirmed by expression increase of mature hepatocyte markers (Cyp3a11, Alb, Ttr, Nrl1h4, Slc2a2, Glul and Nr1h3), ALB and HNF4α as well as by functional assays (Albumin and urea secretion, glycogen storage and LDL uptake). Finally, the authors observed that the proliferation, but not the differentiation capacity, of organoids is profoundly affected by the matrix stiffness, with optimal values in the range of physiological liver (1.3 and 1.7 kPa). The positive action of stiffness was found to rely on the activation of Yes-associated protein 1 (YAP1) since its direct or indirect inhibition negatively affected the organoid formation. Interestingly, the organoid promoting action of stiffness is not dependent on cytoskeletal dynamics but is sensitive to YAP1 phosphorylation by Src family of kinases (SFK) [22].

Another recent study described the generation of human organoids derived by liver biopsies in a chemical nonimmunogenic hydrogel composed by polyisocyanopeptides (PIC). The major advantage of this hydrogel is the thermo-responsive properties that allow the cell recovery by lowering the temperature to 4 °C. Differently from what observed by Sorrentino and colleagues [22], the supplement of hydrogel with RGD was not sufficient to induce organoid proliferation. The authors were able to accomplish the growth of liver organoids by supplementing the PIC matrix with laminin-entactin complex (LEC), the main component of Matrigel. The PIC-LEC matrix supported the growth of human liver organoids with high proliferative rate and the ability to differentiate in mature hepatocytes. The growth of organoids relies upon matrix stiffness though in this case a softer PIC hydrogel (~18 Pa) seems to enhance the organoid expansion. Since LEC has a xenogenic origin, the authors evaluated the organoid growth in PIC hydrogels supplemented with the human recombinant laminin-111. The results obtained are very promising though they did not evaluate the commitment to hepatocyte lineage. Moreover, the use of the PIC matrix was subordinate to an initial culture in Matrigel so rendering the system non completely xenofree and with limits for the clinical application [23]. The current knowledge in xenofree hydrogels is yet limited although the recent promising studies highlight the possibility to explore a wide range of chemicals and ECM proteins in order to accomplish the most suitable combination for the generation of GMP grade organoids for clinical approaches.

## 4. Liver Organoids for Disease Modeling

The possibility to obtain liver organoids from adult tissues and through cell reprogramming of patient-derived cells has led to their application in disease modeling (Figure 2). Patient-derived organoids retain the genetic background of individuals, including the disease-causing mutation allowing their application in personalized medicine and drug efficiency studies. Moreover, by using gene editing, it is possible to introduce pathological mutations in healthy samples in order to assess their role in pathogenesis and responsiveness to therapeutics. Hereafter we describe the advances of organoid culture systems in liver disease modeling (Table 1), regenerative medicine and drug discovery. All these disease models resulted to be compatible with the generation of liver organoids thus paving the way for several further genetic disease applications. Anyways, it is conceivable that some disease genetic mutations might interfere with the organoid development making necessary the employment of inducible systems that allow post-development activation.

### 4.1. Monogenic Diseases

Alagille syndrome (ALGS) is an autosomal dominant genetic disorder mostly caused by a mutation in the NOTCH ligand JAG1 [37]. The hepatic manifestations are characterized by bile duct paucity and chronic cholestasis caused by an impaired biliary differentiation resulting from alterations in NOTCH signaling [38]. The pathological features are mirrored in liver organoids derived by ALGS patient biopsies [24] and mouse models [26] or generated by differentiation of iPSCs derived from ALGS patients [19]. By using CRISPR/Cas9 technology, Guan and colleagues demonstrated the possibility to generate model disease organoids starting from iPSCs derived from healthy donors. By introducing a disease-causing mutation (C829X) in JAG1, they accomplished the generation of liver organoids with ALGS-like phenotype. These authors demonstrated that in iPSC-derived ALGS organoids the development of cholangiocytes and bile ductular structures is impaired and that these disease organoids cannot regenerate secondary organoids [19]. Moreover, they showed that the reversion of the preexisting mutation of ALGS iPSCs restored well-organized structures [19].

Alpha-1 antitrypsin deficiency (A1AT) is caused by mutation in SERPINE1 which results in a deficient production of the serine protease inhibitor alpha-1 antitrypsin. A1AT is mainly produced by the liver and has a crucial role in protecting the lungs from proteolytic damage by regulating the elastase activity of macrophages. Its deficiency results in pulmonary and liver diseases. Particularly, the pathological hepatic manifestations being caused by the accumulation of aberrant protein within the endoplasmic reticulum (ER) resulting in liver damage [39]. Liver organoids of alpha-1 antitrypsin deficiency have been obtained from A1AT patient biopsies. The derived 3D cultures displayed protein aggregates within the cells and a reduced secretion of alpha-1 antitrypsin [12].

Cystic fibrosis (CF) is a genetic disease caused by mutation in the gene encoding the Cystic Fibrosis Transmembrane Conductance Regulator (CFTR). Despite the main pathological manifestations are the persistent lung inflammation and recurrent infections caused by thick mucus, CF is a multisystemic disease that affects other organs. In particular, in the liver the decreased alkalinity and fluidity of bile result in biliary cirrhosis [40]. In cholangiocytes, CFTR is expressed at the apical membrane and responds to hormone stimulation by increasing cAMP intracellular levels resulting in chloride ions efflux in the bile duct lumen. By mimicking the increase of cAMP levels with forskolin (FSK), healthy intestinal organoids internalized fluid in their lumen whereas CFTR patients-derived ones did not respond [41]. The use of this swelling test with iPSC-derived chol-orgs may allow the screening of chemical compounds for the treatment of cystic fibrosis and the association of the disease-causing mutations with the most suitable therapy [26].

Wilson’s disease is a rare genetic disorder associated with mutations in ATP7B, an ATPase involved in copper secretion into the bile. An impairment of ATP7B activity results in intracellular copper accumulation with uncontrolled release in blood circulation. The resulting hepatocyte damage evolves in chronic hepatitis and predisposes patients to develop cirrhosis and hepatocellular carcinoma [42]. Recently, an iPSC-based Wilson disease model was generated from fibroblasts of patients bearing homozygous mutation H1069Q [43], the main mutation found in Caucasian patients. The differentiation of Wilson iPSCs into hepatocytes demonstrated that the copper accumulation is due to a limited amount of H1069Q mutant protein reaching the Golgi complex due to endoplasmic reticulum-associated degradation and in turn to a limited trafficking of ATP7B to the endo/lysosomal compartments in response to Cu overload [44]. A subsequent study demonstrated that ATP7B-H1069Q mutation prevented its Cu2+-induced polarized redistribution to the bile canalicular domain, which could not be reversed by pharmacological folding chaperones [45]. Interestingly, a more recent study demonstrated that, using hepatocyte-like cells derived from Wilson iPSCs, domperidone attenuates degradation of ATP7B-H1069Q [44]. Wilson’s disease 3D model was already developed by using canine liver organoids. However, differently from humans, canine Wilson’s disease is caused by mutations in COMMD1, a scaffold protein that plays a critical role in excretion of copper into the bile. The pathological phenotype of COMMD1-deficient organoids could be reverted by overexpressing the functional protein [36]. In a more recent work, organoids transduced with lentiviral particles to restore COMMD1 were used for autologous transplantation. Despite a low rate of engraftment, the cells persisted in the liver for up to two years [34]. Finally, a system that more closely accounts for the genetic impact that the different ATP7B mutations have on human liver development is still missing. Moreover, considering that the current pharmacological treatments of Wilson’s disease cause serious side effects and exhibit limited efficacy in substantial cohorts of patients [46], patient-derived liver organoids represent an invaluable tool to find new drugs and to develop personalized treatments. Overall a human model of liver organoid is likely to be feasible and beneficial to model Wilson disease.

### 4.2. Steatohepatitis

Following the promising results in the use of organoids for modeling of monogenic liver diseases, 3D structures have been recently used to model more complex acquired diseases. Ouchi and colleagues generated a tri-lineage organoid model from human iPSCs to recapitulate steatosis and steatohepatitis in vitro. The organoids were composed of hepatocyte-, stellate- and Kupffer-like cells. The organoids functional profiles were confirmed by cytochrome p450 activity of hepatocytes, vitamin A storage of stellate cells and LPS response of Kupffer cells. By treating these organoids with oleic acid (OA) they observed a dose-dependent lipid accumulation with induction of inflammation (TNF-α, IL6 and IL8 secretion) and fibrotic phenotype (upregulation of vimentin and α-smooth muscle actin expression, collagen deposition and stiffness increasing). Starting from the patient’s iPSCs, they also generated a 3D model of Wolman disease, a lipid accumulation disorder caused by a defective activity of lysosomal acid lipase [47]. Treatment with FGF19, which activation was described to ameliorate the phenotype [48], suppressed the lipid accumulation, decreased the levels of reactive oxygen species and reduced the organoid stiffening [33].

Lipid accumulation was also observed in murine, feline and canine chol-orgs following treatment with free fatty acids [35]. A drug screening conducted using feline liver organoids, identified two promising candidates that reduced lipid accumulation [49].

Steatohepatitis can also be caused from excessive consumption of alcohol. The disease can evolve in cirrhosis and in a minority of patients in hepatocellular carcinoma [50]. Recently, an in vitro model of alcohol-related liver disease (ALD) was established by human ESC-derived organoids co-cultured with human fetal liver mesenchymal cells. Treatment with ethanol resulted in the induction of ALD features as steatosis, oxidative stress, fibrosis and inflammation [25].

### 4.3. Liver Infections

Although the progress in treatments, liver diseases associated with viral infections remain a global health burden. Human hepatoma cell lines and mouse models do not accurately recapitulate the complex biology of hepatocytes during viral infections. Recent accumulating evidence displayed the feasibility of liver organoids to deep investigate these processes. By expressing hepatocyte-like cell polarity and viral entry factors, 3D structures represent a more accurate model of viral-induced hepatitis. iPSC-derived liver organoids are a suitable model for investigating both Hepatitis B (HBV) and Hepatitis C (HCV) infections allowing cell-to-cell viral transmission [27,28]. Most recently, liver organoids derived from both human adult hepatocytes and cholangiocytes were used to assess the liver tropism of SARS-CoV2 marking the potential role of liver organoids in investigating a broader range of viral infections [32].

### 4.4. Liver Cancer

Primary liver cancer (PLC), including hepatocellular carcinoma (HCC), cholangiocarcinoma (CC) and a combined hepatocellular-cholangiocarcinoma subtype (HCC/CC), is one of the most lethal malignancy worldwide [51]. The research on PLC has been conducted by using 2D culture and mouse models. However, 2D cultures did not mirror the high grade of heterogeneity found in PLCs as the clones with the most beneficial mutations prevail [52]. Patient derived xenografts (PDX), in which cells from patients are transplanted in immunocompromised mice, are more suitable for PLC studies as they recapitulate the features of the original tumors. Despite these models have been successfully used to study resistance mechanisms and drug’s efficacy, they still have several drawbacks such as high costs and time-consuming issues [53].

Liver cancer organoids have been established from surgical resected tissues from patients with HCC, CC and HCC/CC. These tumoroids cultures maintained the genetic pattern of the primary tissue over time and recapitulated the metastatic potential when transplanted in nude mice [29]. Later, HCC and CC organoids were also obtained from needle biopsies [30]. As expected, the establishment of cancer organoids was successful only when the tumor was poorly differentiated and the proliferative index was high, excluding the possibility of deriving tumoroids from patients at early stages of the disease [29,30]. Another limitation in using organoids for cancer modeling is the lack of complexity resembling the mechanisms observed in vivo. The possibility of generating organoids by mixing cancer, stromal and immune cells is still under investigation.

An alternative approach to generate tumoroids is represented by the prime editing of healthy iPSCs or adult stem cells. Clevers’s group demonstrated that by using CRISPR/Cas9 technology to combine loss of function of the tumor suppressor BAP1 with well-known cholangiocarcinoma mutations (TP53, PTEN, SMAD4 and NF1), normal liver organoids acquired malignant features [31]. On the other side, Clevers’s group also showed that tumoroids have a shorter in vitro lifespan when compared to healthy organoids, thus limiting the use of this model for long-term culture applications.

## 5. The Potential Application of Liver Organoids for Regenerative Medicine

Liver disease is a major global health burden affecting millions of people every year. An imbalance of liver function can be related to genetic or environmental causes leading to hepatitis, fibrosis, cirrhosis, cancer, metabolic and autoimmune disorders. Chronic liver disease (CLS) is associated with poor long-term clinical outcome and the only resolutive treatment is represented by transplantation. However, donor organs availability only covers 10% of the global needs resulting in high mortality rate [54]. For this reason, other approaches designed to replace severely injured livers need to be explored.

Liver homeostasis is quite peculiar in comparison with other endodermal organs. It is characterized by a low cellular turnover during epithelial maintenance but when challenged by an injury the liver has a remarkable ability to regenerate. Hyperplasia injury-induced by regenerative signals such as tumor necrosis alpha (TNFα) and interleukin-6 (IL-6) generates complete restoring of the tissue within a week [55]. Moreover, when hepatocyte proliferation is compromised, the liver is still capable of regenerating itself by triggering ductal cell proliferation and differentiation [56].

In recent years, many studies have highlighted the promising potential of applying iPSCs and organoids in regenerative medicine (Figure 2). Liver organoids derived from murine bile ducts were transplanted in a liver failure mouse model. Despite the engraftment percentage being low (1%) the survival rate was significantly increased compared with control mice [11]. Similarly, upon transplantation of biliary duct-derived human organoids in Balb/c nude mice, human Albumin and α-1 antitrypsin were detected in mice serums and persisted for more than 60 days [12]. More recently, an improvement in the engraftment rate, reaching 80%, was observed upon inoculation of mouse primary hepatocyte-derived organoids in Fah-/- mice [14]. Confirming the improving trend in engraftments, organoids derived from human hepatocytes maintain a blood concentration of human Albumin of 10 μg/mL after transplantation in immunodeficient liver damage mouse model [13].

## 6. Liver Organoids as Tools for Drug Discovery

The accumulating successful establishment of liver organoids for disease modeling has aroused much interest on their suitability in drug screening (Figure 2). Shinozawa and colleagues developed a human liver organoid-based assay for the evaluation of drug-induced liver injury (DILI), the main cause for the withdrawal of drugs from the market. They developed a 384 well-based platform referred to as human liver organoids (HLO)-based high throughput toxicity screening (LoT) model and validated 238 marketed drugs by measuring the bile acid transport activity. The platform successfully identified cholestatic drugs showing a very promising attitude in discriminating the output of therapies in presence of genetic variations. By using iPSCs derived HLO with CYP2C9*2 gene variant, bosentan treatment, known to induce liver injury in presence of CYP2C9*2 [57], caused a severe decrease in bile excretion [58].

Cancer drug screening was successfully performed with liver organoids. Indeed, by using this tool, SCH772984, an inhibitor of ERK kinase [59], was identified as a new promising candidate therapeutic agent for CC treatment [29]. Moreover, 3D cultures may represent an innovative tool for validating the mutation specific related response to chemotherapy. Needle biopsy-derived organoids of HCC patients with various etiology and tumor stages displayed a heterogeneous response to sorafenib [30]. A screening of 27 patient-derived cancer organoids with 129 drugs identified a subset of drugs with a universal action on the heterogeneous cultures used [60]. These results indicate the suitability of the system in identifying the most appropriate treatment for personalized therapies (Figure 2).

## 7. Conclusions and Future Perspectives

Liver organoids models have increasingly been incorporated in biomedical research as they present many advantages. They represent a near-physiological architecture recapitulating key hepatic functions and pathophysiology of common liver diseases bearing the genetic background of the human source. Moreover, they can be derived from limited amounts of tissues and genetically manipulated to generate disease models and to reverse the disease-causing mutation for applications in regenerative medicine. Liver organoids can also be cryopreserved in order to generate biobanks of patient derived tissues for wider disease modeling and drug screening studies. Despite the great progress in the field there are still significant limitations. Specifically, the main future challenges in liver organoid field are:(1)Differentiation protocols need to be improved in order to reach higher cell maturation and a correct liver cell type representation, therefore increasing the grade of complexity to more accurately resemble the pathophysiological processes;(2)Development of new bioengineering approaches to guarantee the reproducibility of liver organoid composition and functionality. This goal is fundamental in order to promote liver organoids to drug discovery and pre-clinical testing pipelines;(3)Development of improved synthetic biomaterials to obtain an animal grade free culturing system for future safe applications in regenerative medicine.

A final perspective application for liver organoids is indeed their use to regenerate damaged liver tissue in patients. Transplantation of liver organoids in animal models have displayed promising results [12,13,14] however protocols need to be further improved in order to increase the rate of engraftment and promote the connection with host blood vessels for oxygen and nutrients supply [16,17]. Overcoming these issues is the final step to allow tissue engineering to become a reality in liver disease treatment.

## Figures and Tables

**Figure 1 biology-10-00835-f001:**
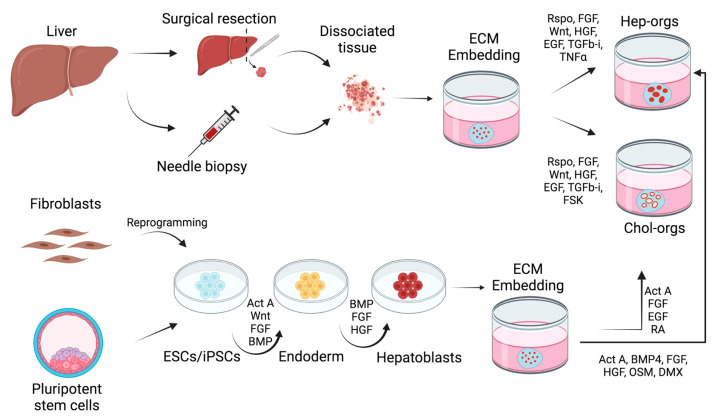
Schematic representation of the generation of liver organoids. Liver organoids can be derived from different sources. Tissue resident cells derived from surgical resection and needle biopsies of patients can be stimulated to form hep-orgs and chol-orgs following incubation with a defined cocktail of growth factors supplemented in medium. PSCs and iPSCs require a 3-stage differentiation protocol to generate hepatoblasts that are subsequently embedded in ECM (Matrigel or defined proteins) to promote organoids formation. Signaling pathways modulated to enhance the generation of organoids are indicated. Act A, activin A; BMP4, bone morphogenetic protein 4; DMX, dexamethasone; ECM, extracellular matrix; EGF, epidermal growth factor; ESCs, embryonic stem cells; FGF, fibroblast growth factor; FSK, forskolin; HGF, hepatocyte growth factor; iPSCs, induced pluripotent stem cells; OSM, oncostatin M; RA, retinoic acid; Rspo, R-Spondin; TGFb-I, transforming growth factor beta inhibitor; TNFα, tumor necrosis factor alpha.

**Figure 2 biology-10-00835-f002:**
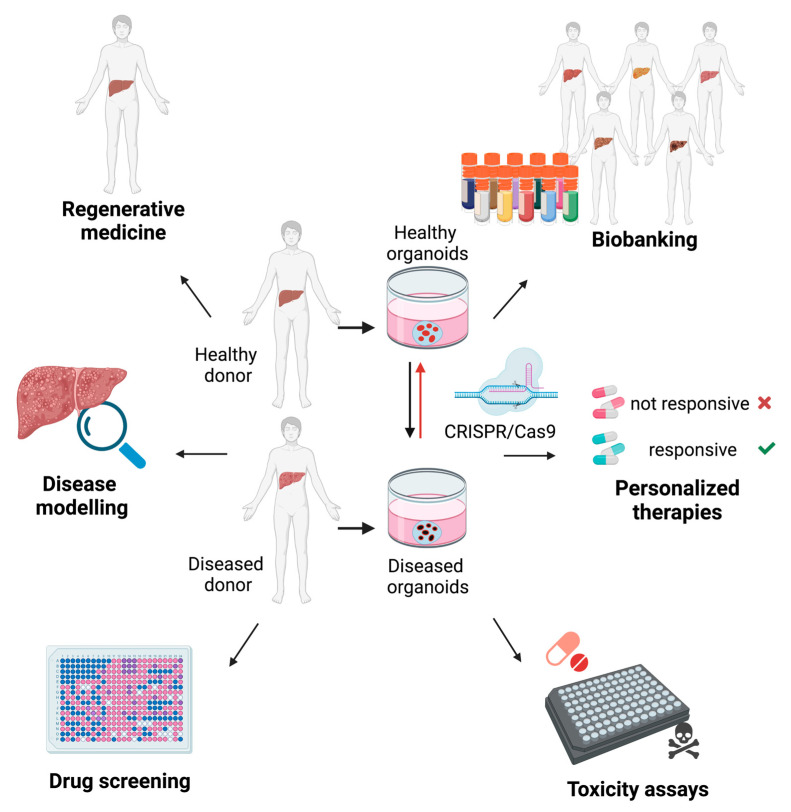
Applications of liver organoids. Healthy and diseased patient-derived organoids represent a precious model to investigate the physiopathology of the liver. Liver organoids harboring mutations for liver diseases studies can be additionally generated by CRISPR/Cas9 gene editing. Liver organoids cover a broad range of biomedical applications including personalized medicine to identify patient-specific responses to drugs. Moreover, cryopreservation enables the establishment of biobanks that can be used for large-scale studies as drug screening, toxicity assays and regenerative medicine.ta.

**Table 1 biology-10-00835-t001:** Liver organoids application as disease modeling.

Disease	Species	Source	References
Alagille syndrome	Human	Biopsy	[12]
		iPSCs	[19]
	Mouse	Adult tissue	[24]
Alpha-1 antitrypsin deficiency	Human	Biopsy	[12]
Alcohol-related steatohepatitis	Human	ESCs	[25]
Cystic fibrosis	Human	iPSCs	[26]
HBV infection	Human	iPSCs	[27]
HCV infection	Human	iPSCs	[28]
Liver cancer	Human	Biopsy	[29,30]
	Human	Adult tissue (GE)	[31]
Sars-CoV-2 infection	Human	PSCs	[32]
Steatosis and steatohepatitis	Human	iPSCs	[33]
	Cat	Adult tissue	[34,35]
Wilson’s disease	Dog	Adult tissue	[34,36]

iPSCs, induced pluripotent stem cells; ESCs, embryonic stem cells; PSCs, pluripotent stem cells; GE, genetically engineered.

## Data Availability

Not applicable.

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
