# Peer review of "Liver Organoids: Updates on Disease Modeling and Biomedical Applications"

_biology, 2021, doi:10.3390/biology10090835_

Round 1

Reviewer 1 Report

The authors present a great and up to date review on the status of liver organoids in modeling human disease. Several aspects are presented including an overview on the history and progress on organoid development from various sources including iPSC, adult stem cells, patient derived biopsies, etc. They also discuss the state of scaffold research comparing Matrigel systems to other synthetic biopolymers. Several examples were provided to demonstrate the avenues for disease modeling ranging from monogenic liver disorders to cancers. Lastly, potential for 3D organoids for regenerative therapy and drug discovery applications was discussed.

Minor concerns:

  • Please proof-read the manuscript. e.g., Title for subsection 4.1 was in bold unlike others. A few other spelling/ sentence formations found through the document.
  • Despite significant progress, there are still several limitations to the application of organoids and I would like to see these challenges discussed in such a review.

Author Response

We are grateful for Reviewer#1 comments.

As requested we revised all the main text for spelling and grammar and expanded the discussion of liver organoid limitations in the discussion section.

Reviewer 2 Report

Dr Carmen Caiazza et al have updated the information regarding liver organoid development, in terms of disease modeling and biomedical application.

The review is well written and I enjoy reading it. Here are some comments that might help to improve the manuscript.

  1. In figure 1 and the text, please describe a little more regarding the development and the compositions of extracellular matrix (ECM) used. As in the latter sections, you have tried to compare between the ECM and other new products.
  2. Please comment on the feasibility of the disease modeling systems listed in Table 1. Can these systems be maintained indefinitely? Will the organoids be lost after some periods of time?
  3. What is the reason that we cannot obtain human Wilson’s disease 3D model?
  4. Line 314: Change “hepatocarcinoma” to “hepatocellular carcinoma”.
  5. Liver cancer organoids: Any information regarding the changes of phenotypes or genetics after several passages of the cancer tumoroids?

Author Response

We are grateful to Reviewer#2 for the comments. We addressed all the raised points by modifying the main text in the relative sections.
